# POH1 deubiquitinates pro-interleukin-1β and restricts inflammasome activity

Li Zhang[1], Yun Liu[1], Boshi Wang[1], Guiqin Xu[1], Zhaojuan Yang[1], Ming Tang[1], Aihui Ma[1], Tiantian Jing[1], Xiaoli Xu[1], Xiaoren Zhang[2] & Yongzhong Liu [1]

Inflammasome activation is essential for host defence against invading pathogens, but is also involved in various forms of inflammatory diseases. The processes that control inflammasome activity are thus important for averting excessive immune responses and tissue damage. Here we show that the deubiquitinase POH1 negatively regulates the immune response triggered by inflammasome activation. POH1 deficiency in macrophages enhances mature IL-1β production without significant alterations in inflammasome priming and ASC-caspase-1 activation. In WT macrophages, POH1 interacts with and deubiquitinates pro-IL-1β by decreasing the K63-linked polyubiquitin chains, as well as decreases the efficacy of pro-IL-1β cleavage. Furthermore, myeloid cell-specific deletion of POH1 aggravates lipopolysaccharide-induced systemic inflammation and alum-induced peritonitis inflammatory responses in vivo. Our study thereby reveals that POH1-mediated deubiquitination of pro-IL-1β is an important regulatory event that restrains inflammatory responses for the maintenance of immune homeostasis.

---

[1] State Key Laboratory of Oncogenes and Related Genes, Shanghai Cancer Institute, Renji Hospital, Shanghai Jiao Tong University School of Medicine, 200032 Shanghai, China. [2] The Key Laboratory of Stem Cell Biology, Institute of Health Sciences, Shanghai Jiao Tong University School of Medicine (SJTUSM) & Shanghai Institutes for Biological Sciences (SIBS), Chinese Academy of Sciences (CAS), 200025 Shanghai, China. These authors contributed equally: Li Zhang, Yun Liu. Correspondence and requests for materials should be addressed to Y.L. (email: liuyzg@shsci.org)

nflammasome activation initiates innate immune responses through triggering the maturation and secretion of the pro-inflammatory cytokines interleukin-1β (IL-1β) and IL-18[1–3]. Upon activation of pattern recognition receptors (PRRs) by pathogen-associated molecular patterns (PAMPs) or damage-associated molecular patterns (DAMPs), the expression of the inactive pro-inflammatory cytokine precursors and inflammasome sensors are induced in a nuclear factor-κB (NF-κB) activity-dependent manner[1–4]. Canonical inflammasome complexes are then assembled by activation of cytosolic receptors that contain an evolutionarily conserved PYRIN domain (PYD) or caspase recruitment domain (CARD), as well as by recruitment of ASC (apoptotic speck-like protein containing a CARD) or CAD-DINAL and pro-caspase-1[3–5]. Formation of the macromolecular complexes induces a proteolytic cascade that is initiated by caspase-1, followed by maturation and secretion of IL-1β and IL-18, which activate inflammatory responses[1–5].

Although inflammasome activation is critical for pathogen clearance and tissue injury recovery, inappropriate inflammasome activation leads to harmful effects and is involved in a variety of inflammatory disorders[2,6–8]. Tight regulation of inflammasome activation and maturation of the pro-inflammatory cytokine IL-1β are therefore critical for avoiding excessive responses and preserving homeostasis[1,2,7]. However, the precise mechanism underlying the negative control of the immune response evoked by inflammasomes remains poorly understood.

Post-translational modification of inflammasome components by ubiquitin is critical for negative regulation of inflammasome activation[9]. Ubiquitination of NLRP3, ASC (apoptotic speck-like protein containing a CARD) and other inflammasome components, mostly leading to degradation of these proteins, regulates several key nodes in the endogenous regulatory networks[9–14]. The E3 ubiquitin ligases, such as MARCH7, FBXL2 and TRIM31, have been shown to promote K48-linked polyubiquitination and degradation of NLRP3[9,10,15,16]. In addition, K63-linked polyubiquitination of the ASC inflammasome leads to a negative feedback, autophagy-related process that destroys the AIM2 inflammasome[17]. Importantly, downstream IL-1β is also subjected to ubiquitination-mediated regulation that controls the cleavage efficiency or turnover of pro-IL-1β[18,19]. Given that the regulation is effective and relatively distal in the signalling of inflammasome-mediated IL-1β activation, the process of pro-IL-1β deubiquitination may be broadly implicated in regulating the activities of different inflammasomes. Currently, whether novel mechanisms related to the regulation of pro-IL-1β ubiquitination exist is still unknown.

POH1/rpn11/PSMD14 is a K63-specific deubiquitinase within the 19S particle lid of the proteasome and plays critical roles in regulating multiple signalling pathways, including maintenance of protein stability, double-strand DNA break repairs, embryonic stem cell differentiation and cellular proliferation[20–24]. We previously showed that hyper-activated POH1-E2F1 regulation promotes the development of liver cancer[20]. A recent study reported that POH1 intensifies TRIM21-induced activation of NF-κB signalling and subsequently promotes IL-6 and tumour necrosis factor α (TNFα) expression at their transcriptional levels in murine embryonic fibroblast cells upon AdV/IgG stimulation[25]. However, the role of POH1 in immune responses evoked by inflammasome activation is still undefined.

Here, we report that the deubiquitinase POH1 is a negative regulator of immune responses that are triggered by inflammasome activation. POH1 decreases the K63-linked polyubiquitination of pro-IL-1β and restricts its processing. Furthermore, POH1 upregulation by toll-like receptor (TLR) stimulation suggests a POH1-mediated negative feedback control for inflammasome-dependent inflammation. The importance of

the POH1 regulation of pro-IL-1β cleavage is exemplified by the occurrence of excessive inflammation in myeloid POH1-deficient mice upon lipopolysaccharide (LPS) or alum treatment. These results reveal that the POH1-mediated deubiquitination of pro-IL-1β is crucially required to avert inappropriate inflammatory responses and preserve homeostasis.

## Results

**POH1 in myeloid cells protects from excessive inflammation.** To investigate whether POH1 regulates the innate immune response, we first generated mice with specific deletion of POH1 in myeloid cells by crossing using the *LysM-Cre+* mice and the *Poh1*[F/F] mice[20]. No apparent phenotypical difference was found in mice with a heterozygous deletion of *Poh1* (*LysM-Cre+/ Poh1*[F/+], hereafter termed *Poh1*[Δ/+]) compared with *Poh1*-wild-type littermates (*LysM-Cre+/ Poh1*[+/+], *Poh1*[+/+]). By contrast, mice with a homozygous deletion of *Poh1* (*LysM-Cre+/ Poh1*[F/F], *Poh1*[Δ/Δ]) developed splenomegaly and had an increase in the number of splenocytes beginning at 8–12 weeks of age (Supplementary Fig. 1a). Further analysis of splenic cell subpopulations demonstrated that the absolute numbers of CD11b[+] F4/80[+] macrophages, CD11b[+] Ly6G[+] neutrophils and CD11b[+] Ly6C[+] monocytes from *Poh1*[Δ/Δ] mice were increased compared with those from control *Poh1*[Δ/+] littermates (Supplementary Fig. 1b). Furthermore, the total numbers of splenic CD4[+] or CD8[+] T cells were comparable between *Poh1*[Δ/+] and *Poh1*[Δ/Δ] mice, but the counts of B220[+] B cells were augmented in myeloid POH1-deficient spleens (Supplementary Fig. 1b). In addition, the total number of neutrophils was increased in POH1-deficient bone marrow (Supplementary Fig. 1c). Intriguingly, increased serum levels of IL-1β were also observed in *Poh1*[Δ/Δ] mice compared with control mice (Supplementary Fig. 1d). To further investigate the role of POH1 in endotoxin-induced inflammation, we examined the responses of *Poh1*[Δ/Δ] mice and their littermates (*Poh1*[+/+] or *Poh1*[Δ/+] mice) upon LPS challenge. Although approximately 57.2% of *Poh1*[+/+] and 78.6% of *Poh1*[Δ/+] mice survived to LPS injection, all *Poh1*[Δ/Δ] littermates succumbed to the same treatment (Fig. 1a). The observation indicates that POH1 has a critical role in regulating pathogenic inflammatory responses.

NF-κB signalling is known to be required for TLR4-triggered transcriptional activation of inflammation-related cytokines, including IL-6, TNFα and pro-IL-1β[26,27]. NLRP3 inflammasome activation after LPS priming is responsible for IL-1β maturation[4,12]. We next tested whether POH1 deficiency causes alterations in the expression and secretion of these cytokines in mice upon LPS treatment. Compared with control mice, *Poh1*[Δ/Δ] mice had a substantial increase in serum IL-1β concentrations, whereas the serum levels of IL-6 or TNFα were not significantly altered (Figs. 1b–d). Consistently, there were no pronounced differences in the levels of IL-6, TNFα and IL-1β transcripts in lung tissues between the two groups (Fig. 1e). To corroborate the results, *Poh1*[Δ/Δ] and control *Poh1*[Δ/+] bone marrow–derived macrophages (BMDMs) stimulated with LPS were subjected to a measurement of transcriptional expression of IL-1β, IL-6 or TNFα, and the results confirmed that the abundance of these transcripts was not markedly regulated by POH1 (Supplementary Fig. 2a-c). Thus, these results suggest that POH1 is an important negative regulator of IL-1β production, and that the regulation may occur in the stage of NLRP3 inflammasome activation or during pro-IL-1β processing.

To better determine the contribution of POH1 to the regulation of NLRP3-induced inflammation, we investigated the function of POH1 in the alum-induced, NLRP3 inflammasome-dependent peritonitis mouse model. Consistently, *Poh1*[Δ/Δ] mice had increased amounts of IL-1β in cleared peritoneal lavage fluids

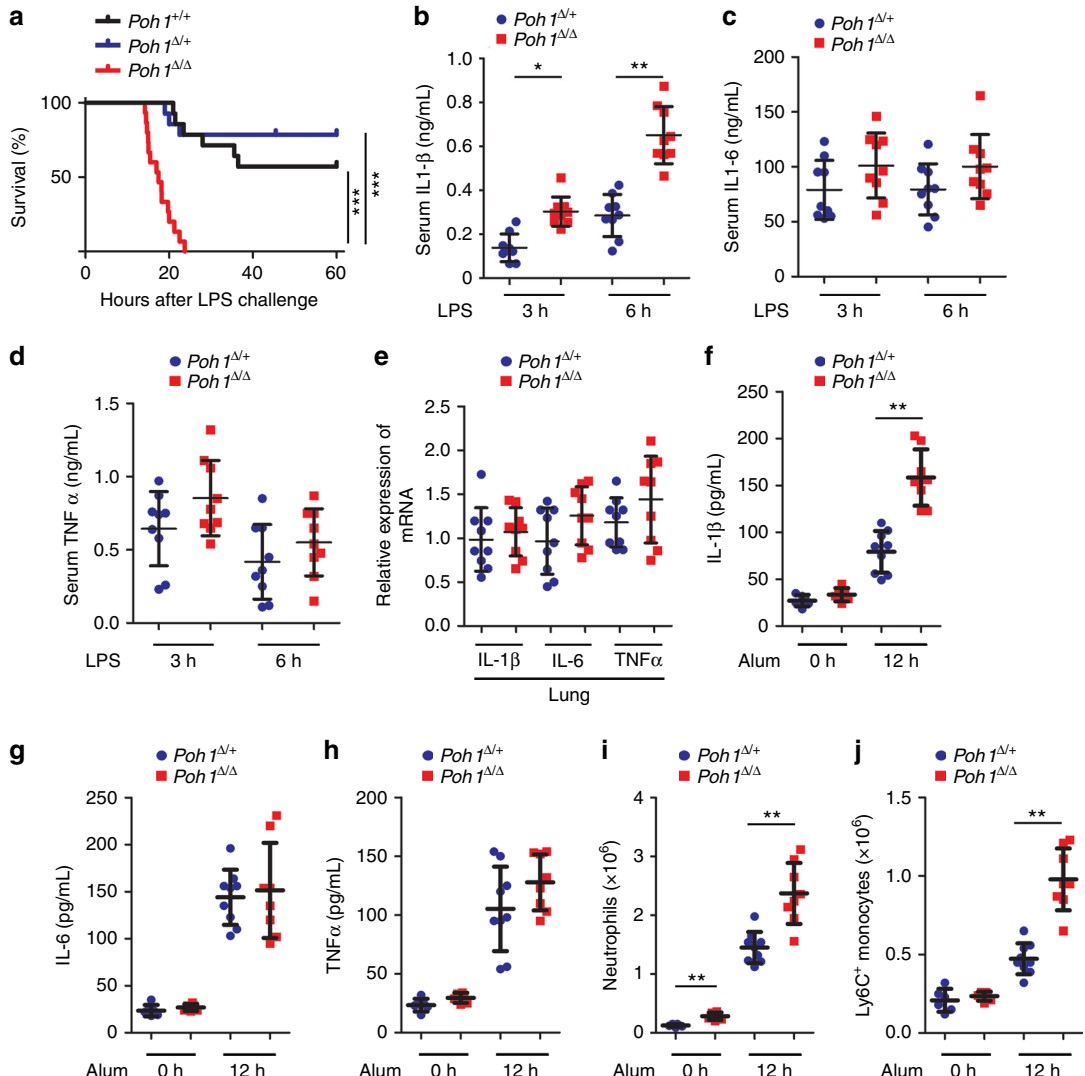

**Fig. 1** Myeloid POH1 deficiency accelerates inflammation in vivo. **a** $Poh1^{+/+}$ ($n = 14$), $Poh1^{\Delta/+}$ ($n = 14$) and $Poh1^{\Delta/\Delta}$ ($n = 15$) mice were intraperitoneally (i. p.) injected with LPS (40 mg/kg). Survival was monitored for up to 60 h, ***$p < 0.001$ (asymmetrical log-rank Mantel–Cox survival test). **b–d** ELISA quantification of the (**b**) IL-1β, (**c**) IL-6 and (**d**) TNFα levels in the serum of $Poh1^{\Delta/+}$ and $Poh1^{\Delta/\Delta}$ mice upon LPS challenge ($n = 9$ per group). **e** Lungs of $Poh1^{\Delta/+}$ and $Poh1^{\Delta/\Delta}$ mice were harvested 6 h post-LPS injection, and then the transcriptional levels of IL-1β, IL-6 and TNFα were measured by RT-PCR ($n = 9$ per group). **f–j** $Poh1^{\Delta/+}$ and $Poh1^{\Delta/\Delta}$ mice were i.p. injected with ($n = 8$–9 per group) or without ($n = 6$ per group) 40 mg/kg of alum crystals, and then peritoneal fluids were elicited 12 h after the treatment and subjected to analysis of **f** IL-1β, **g** IL-6 and **h** TNFα by ELISA; absolute numbers of **i** CD11b+ Ly6G+ neutrophils or **j** CD11b+ Ly6C+ monocytes recruited to the peritoneum were measured. Data are pooled from **a** three or **b–j** two independent experiments (mean ± s.e.m. in **b–h**; mean ± s.d. in **i, j**), *$p < 0.05$, **$p < 0.01$ (two-tailed Student's $t$-test)

compared with $Poh1^{\Delta/+}$ mice upon alum challenge (Fig. 1f). However, inflammasome-independent release of IL-6 or TNFα was not significantly altered by POH1 deletion (Figs. 1g, h). Intraperitoneal (i.p.) injection of alum crystals usually causes IL-1β-dependent neutrophil infiltration[10,28]. As expected, we observed potent influxes of neutrophils and Ly6C+ monocytes in the peritoneal cavities of alum-treated $Poh1^{\Delta/+}$ mice. The responses, however, were strongly enhanced in $Poh1^{\Delta/\Delta}$ mice (Figs. 1i, j). Taken together, our experiments demonstrate that POH1 deficiency in myeloid cells significantly aggravates the NLRP3 inflammasome responses in vivo.

**POH1 expression is upregulated by the TLR activators.** The negative role of POH1 in regulating LPS-induced septic shock prompted us to examine the expression pattern of POH1 in macrophages during TLR activation. We found that POH1 was upregulated in BMDMs at different time points after LPS or

poly(I:C) stimulation (Fig. 2a, Supplementary Fig. 2d). The TLR9 ligand CpG or IL-1β, however, failed to induce POH1 expression in BMDMs (Fig. 2a, Supplementary Fig. 2d). Furthermore, the remarked increases in POH1 expression were observed as early as 1 h after LPS stimulation and maintained for at least for 24 h in BMDMs (Fig. 2b, Supplementary Fig. 2e). A similar tendency in POH1 induction also occurred in poly(I: C)-stimulated BMDMs (Fig. 2c, Supplementary Fig. 2f). The rapid induction of POH1 in macrophages suggests that a post-transcriptional mechanism is involved in the upregulation of POH1 by LPS or poly(I:C). Indeed, the transcriptional levels of POH1 were barely changed in cells during the stimulations (Figs. 2d, e). Collectively, the function of POH1 to prevent excessive immune responses and POH1 upregulation in the stage of inflammasome priming reinforce the physiological implication of POH1 in the negative regulation of inflammation evoked by inflammasome activation.

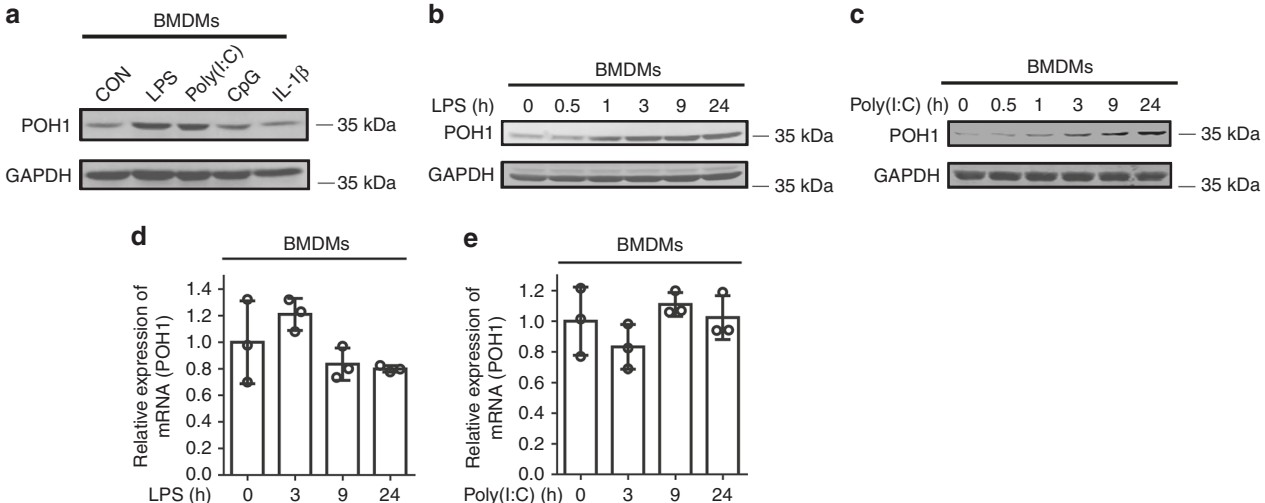

**Fig. 2** POH1 expression is upregulated in macrophages by TLR3/4 activation. **a** BMDMs from WT mice were stimulated with LPS (100 ng/mL), poly(I:C) (10 μg/mL), CpG (5 μM) or IL-1β (10 ng/mL) for 9 h and analysed for expression of POH1 by immunoblot. **b, c** Immunoblot analysis of POH1 expression from BMDMs treated with **b** LPS or (**c**) poly(I:C) as indicated. **d, e** RT-PCR analysis of POH1 expression in BMDMs in response to **d** LPS or **e** poly(I:C) stimulation. Similar results were obtained from three independent experiments. The results represent the mean ± s.d. of three independent sets of experiments

**POH1 negatively regulates IL-1β production.** As previously mentioned, POH1 was capable of restricting IL-1β production in immune responses triggered by NLRP3 inflammasome activation. However, POH1 deficiency did not cause a remarkable change in the expression of the NLRP3 inflammasome components, including NLRP3, ASC and pro-caspase-1, as well as of pro-IL-1β in BMDMs upon LPS stimulation (Supplementary Fig. 2g). To further ascertain whether POH1 is a regulator of inflammasome-induced IL-1β generation, we primed BMDMs from $Poh1^{\Delta/+}$ and $Poh1^{\Delta/\Delta}$ mice with LPS and subsequently treated them with the NLRP3 inflammasome activators ATP or nigericin. As expected, POH1-deficient BMDMs released more IL-1β than control cells, whereas the alterations in the amounts of IL-6 and TNFα were marginal (Figs. 3a–c). Intriguingly, upregulation of IL-1β release due to POH deficiency was not restricted to NLRP3 inflammasome-activated macrophages. We found that POH1 deficiency also caused significant increases in IL-1β secretion in macrophages stimulated by either the agonist of AIM2 inflammasome, poly(dA:dT), or by the agonist of NLRC4 inflammasome, flagellin (Fig. 3a). Consistently, POH1 overexpression greatly inhibited IL-1β secretion in LPS-primed cells upon stimulation of ATP, nigericin, poly(dA:dT), flagellin (Fig. 3d), whereas no significant differences were found in the production of IL-6 or TNFα in these BMDMs (Figs. 3e, f). Collectively, these results indicate that POH1 in macrophages inhibits IL-1β activation upon inflammasome activation.

Inflammasome-dependent caspase-1 activation is required for the maturation of IL-1β[1]. We therefore investigated whether POH1 regulates inflammasome-induced pro-caspase-1 cleavage. Unexpectedly, upon treatment with the inflammasome activators, POH1 deficiency did not lead to an increase in pro-caspase-1 cleavage in LPS-primed BMDMs, but enhanced cleavage of pro-IL-1β, as measured by the mature forms of pro-caspase-1 and pro-IL-1β (Fig. 3g, Supplementary Fig. 3a, b). Consistently, cleavage of pro-IL-1β but not of pro-caspase-1 was attenuated in POH1-overexpressing macrophages compared with control cells in the presence of the activators (Fig. 3h).

ASC cytoplasmic foci formation is an important event that occurs upstream of pro-caspase-1 cleavage and is an additional marker to evaluate inflammasome activation[1,4,12]. We next examined the efficiency of ASC cytoplasmic foci formation in POH1-deficient and control BMDMs. After LPS priming and subsequent stimulation

with the inflammasome agonists, endogenous ASC in macrophages was visualized by immunofluorescent staining. The results showed that the percentages of macrophages containing the ASC pyroptosome were comparable between POH1-deficient BMDMs and control $Poh1^{\Delta/+}$ BMDMs (Figs. 4a, b). ASC pyroptosomes can be isolated from total cell lysate using a non-reversible crosslinking agent (disuccinimidyl suberate (DSS)) to determine the extent of ASC oligomerization[29,30]. To further confirm our results, we examined ASC polymerization and found that the amount of polymerized ASC in control BMDMs was similar to that formed in POH1-deficient BMDMs upon agonist treatment (Fig. 4c). Furthermore, POH1 overexpression had no effect on ASC cytoplasmic foci formation or ASC oligomerization in BMDMs (Figs. 4d–f), indicating that POH1 expression does not have an appreciable impact on inflammasome assembly. In addition, we reconstituted inflammasome components in HEK293T cells by transient expression of ASC, pro-caspase-1 and pro-IL-1β, together with or without POH1, and examined the activation of pro-caspase-1 and pro-IL-β. In accordance with the results (Figs. 3g, h), cleavage of pro-IL-1β but not of pro-caspase-1 was inhibited by POH1 overexpression in a dose-dependent manner (Fig. 4g). As expected, ectopic POH1 expression also substantially inhibited IL-1β secretion (Fig. 4h). Meanwhile, we found that the amounts of IL-18 in supernatants of POH1-deficient BMDMs were not significantly changed compared with their controls (Supplementary Fig. 4a). Consistently, POH1 deficiency did not significantly alter the levels of cleaved gasdermin D (GSDMD) in cell lysates of BMDM stimulated with these agonists (Supplementary Fig. 4b). Similar results were also obtained in HEK293T cells reconstituted with ASC, pro-caspase-1, and their substrates, IL-18 or GSDMD, together with or without POH1; POH1 overexpression had little effect on the cleavage of IL-18 or GSDMD (Supplementary Fig. 4c, d). In addition, no significant difference in the inflammasome-dependent pyroptosis was found between POH1-deficient BMDMs and control cells (Supplementary Fig. 4e). Collectively, these results reveal a critical role of POH1 in the regulation of pro-IL-1β processing and demonstrate that the restriction by POH1 is not through modulating inflammasome assembly.

**POH1 interacts with pro-IL-1β.** It has been demonstrated that POH1 interacts with and functionally modifies multiple

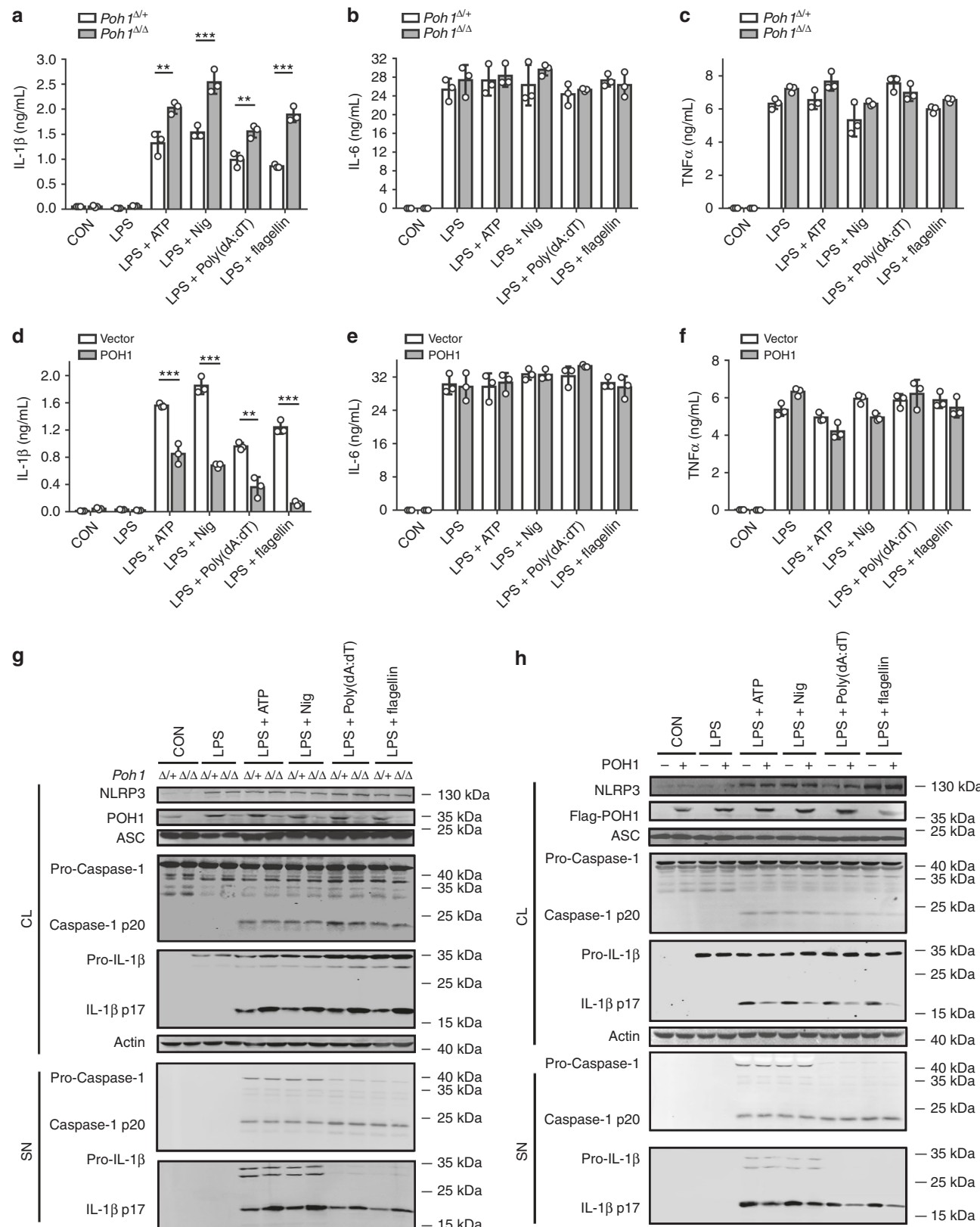

substrates in different contexts[20,21,25]. We next extended our study to test the possibility that POH1 can interact with inflammasome complex components, especially pro-IL-1β, under physiological conditions. We immunoprecipitated POH1 from BMDMs pretreated with LPS, and found that POH1 could interact with pro-IL-1β, pro-caspase-1 and NLRP3 (Fig. 5a). To

confirm our results, we co-expressed POH1 with pro-IL-1β, NLRP3, ASC or pro-caspase-1 in HEK293T cells, which do not express endogenous NLRP3 inflammasome components. After immunoprecipitation with a Flag antibody, we detected interactions of POH1 with pro-IL-1β and pro-caspase-1, but not with NLRP3 and ASC (Figs. 5b, c). In addition, we

**Fig. 3** POH1 inhibits inflammasome-induced pro-IL-1β processing. **a–c** BMDMs from $Poh1^{\Delta/+}$ and $Poh1^{\Delta/\Delta}$ mice were primed with LPS for 12–14 h and then stimulated with ATP (0.5 h), nigericin (Nig, 0.5 h), poly(dA:dT) (5 h) or flagellin (5 h) as indicated, **a** IL-1β, **b** IL-6 and **c** TNFα levels in supernatants were analysed by ELISA. **d–f** BMDMs lentivirally transduced with control (Vector) or POH1 were stimulated as in **a**, **d** IL-1β, **e** IL-6 and **f** TNFα levels in supernatants were analysed by ELISA. **g, h** BMDMs **g** derived from $Poh1^{\Delta/+}$ and $Poh1^{\Delta/\Delta}$ mice or **h** infected by indicated lentiviruses were treated as in **a**, the cell lysates (CL) and supernatants (SN) were collected and immunoblotted with the indicated antibodies. Similar results were obtained from three independent experiments. The results represent the mean ± s.d. of three independent sets of experiments. **$p < 0.01$, ***$p < 0.001$ (two-tailed Student's $t$-test)

immunoprecipitated endogenous pro-IL-1β or pro-caspase-1 in BMDMs stimulated with LPS, and the immunoprecipitates were immunoblotted for POH1. The results showed that endogenous pro-IL-1β and pro-caspase-1 could interact with POH1 (Supplementary Fig. 5a). As previously reported, K63-linked polyubiquitination of pro-caspase-1 is critical for its activity[13]. We therefore co-expressed pro-IL-1β, pro-caspase-1 and ASC with HA-K48-ubiquitin (only for K48-linked conjugation) or HA-K63-ubiquitin (only for K63-linked conjugation) in HEK293T cells and immunoprecipitated pro-caspase-1 under highly stringent conditions. By probing with an anti-hemagglutinin (HA) antibody, we found that pro-caspase-1 was modified with K48-linked and K63-linked polyubiquitin chains (Fig. 5d middle panel, Supplementary Fig. 5b). Interestingly, overexpression of POH1 did not have appreciable effects on these modifications (Fig. 5d middle panel, Supplementary Fig. 5b). To identify which domain of POH1 interacts with pro-IL-1β, we generated five Flag-tagged POH1 truncated mutants, namely, POH1 ΔJAMM, the mutant form of POH1 with a deletion of the JAMM motif; POH1 ΔC, the mutant form of POH1 without amino acids 181–310; POH1 ΔN, N-terminal deleted POH1 that lacks amino acids 1–180; POH1-M (amino acids 141–230) and POH1-C (amino acids 231–310) (Supplementary Fig. 5c). We found that while ΔN-, ΔJAMM, ΔC-POH1 and POH1-M mutants were associated with pro-IL-1β, albeit at different efficiencies, POH1-C failed to interact with pro-IL-1β (Supplementary Fig. 5d). Collectively, these results show that POH1 interacts with pro-IL-1β prior to inflammasome activation.

**POH1 inhibits K63-linked polyubiquitination of pro-IL-1β.** Unrestricted ubiquitination of pro-IL-1β enhances its processing and causes excessive NLRP3 inflammasome activity[19]. Intrigued by the physical interaction that occurs between POH1 and pro-IL-1β, we first sought to examine whether POH1 regulates pro-IL-1β ubiquitination. The results showed that POH1 efficiently reduced K63-linked, but not K48-linked, ubiquitination on pro-IL-1β (Fig. 5d right panel, Supplementary Fig. 6a), consistent with previous studies characterizing POH1 as a K63-specific deubiquitinase[23,24]. To further confirm the POH1-induced deubiquitination of pro-IL-1β under physiological conditions, we treated $Poh1^{\Delta/\Delta}$ and control $Poh1^{\Delta/+}$ BMDMs with LPS or LPS plus the NLRP3, AIM2 and NLRC4 inflammasome agonists. Although endogenous pro-IL-1β was ubiquitinated with K48-linked and K63-linked chains upon stimulation with the inflammasome agonists (Fig. 5e, Supplementary Fig. 6b, c), the amount of K63-linked but not K48-linked polyubiquitin chains on pro-IL-1β was much higher in $Poh1^{\Delta/\Delta}$ BMDMs than in $Poh1^{\Delta/+}$ BMDMs (Fig. 5e, Supplementary Fig. 6b, c).

To further investigate whether POH1 regulates pro-IL-1β processing through its deubiquitinase activity, we generated a POH1 mutant, POH1-H113Q, which lacks deubiquitinating enzymatic activity[20,31–34]. In cells transduced with plasmids expressing pro-IL-1β, ASC, pro-caspase-1 and HA-K63-ubiquitin, co-expression of wild-type (WT) POH1 profoundly attenuated K63-linked ubiquitination on pro-IL-1β, whereas POH1-H113Q expression did not affect modification of pro-IL-1β by K63-linked polyubiquitination (Fig. 5f, Supplementary Fig. 6d). Clearly, in

stark contrast with WT POH1, POH1-H113Q failed to inhibit pro-IL-1β cleavage (Supplementary Fig. 6e). Thus, our results indicate that the deubiquitinase activity of POH1 is critically required for POH1-mediated negative regulation of pro-IL-1β processing. K133, an evolutionarily conserved lysine in murine pro-IL-1β, is considered to be responsible for K63-linked ubiquitination of pro-IL-1β under physiological conditions[19]. Mutation of K133 on pro-IL-1β has a functional consequence with regard to the efficiency of pro-IL-1β processing[19]. In our settings, the importance of K133 in IL-1β activation was also evident by significant decreases in the cleavage of pro-IL-1β and secretion of the mature form in cells reconstituted with the components for pro-IL-1β processing (Figs. 6a, b). Unexpectedly, co-expression of POH1 was still capable of inhibiting the processing of pro-IL-1β with the K133R mutation (Figs. 6a, b). We utilized mass spectrometry analysis and found that five lysine residues of murine pro-IL-1β (K30, K133, K205, K209 and K247) were ubiquitinated (Supplementary Fig. 7a–e). We therefore generated five murine pro-IL-1β mutants by replacing these lysine residues with arginine: K30R, K205R/K209R, K247R, K30R/K205R/K209R/K247R (M4) and K30R/K133R/K205R/K209R/K247R (M5) (Fig. 6c). Remarkably, K63-linked ubiquitination on the mutants of K133R, K247R and those containing K133R and/or K247R (M4, M5) were decreased in relative to that on WT pro-IL-1β. Consistently, the mutants with K133R and/or K247R were less efficient in the generation of activated IL-1β and cleavage of pro-IL-1β (Figs. 6d–f, Supplementary Fig. 8a). Moreover, the mutants K247R, K133R and M4, which retained a considerable modification of K63-linked ubiquitin, were vulnerable to POH1-mediated deubiquitination (Supplementary Fig. 8b); the inhibitory effects of POH1 on IL-1β release were observed in these mutants (Supplementary Fig. 8c). Collectively, these data suggest the importance of deubiquitinating pro-IL-1β at K133 and K247 in POH1-mediated negative regulation of pro-IL-1β processing.

## Discussion

As a pro-inflammatory cytokine, IL-1β functions as a major effector of the inflammasome signalling pathway that links inflammasome activation to multiple downstream events in the innate immune response[1,7]. Here, we demonstrate that POH1 plays a critical role in the restriction of IL-1β maturation, for which the deubiquitinating enzymatic activity of POH1 is required. Remarkably, POH1 upregulation in macrophages upon TLR activation underscores the physiological significance of POH1 in the self-control of inflammasome activity. The exaggerated inflammation in myeloid cell-POH1-deficient mice treated with the NLRP3 inflammasome activators highlights the importance of POH1-mediated negative regulation. Our study identifies an important function for the deubiquitinase POH1 in the regulation of the innate immune response.

Initiation of inflammasome activation requires priming signalling that turns on transcription of inflammatory cytokines and components of the inflammasome[1,3,14]. The data collected from our in vitro experiments and mouse models showed that POH1 deficiency did not compromise transcriptional activation of the genes that are essential for inflammasome-dependent inflammations. In primed macrophages with or without POH1 deficiency,

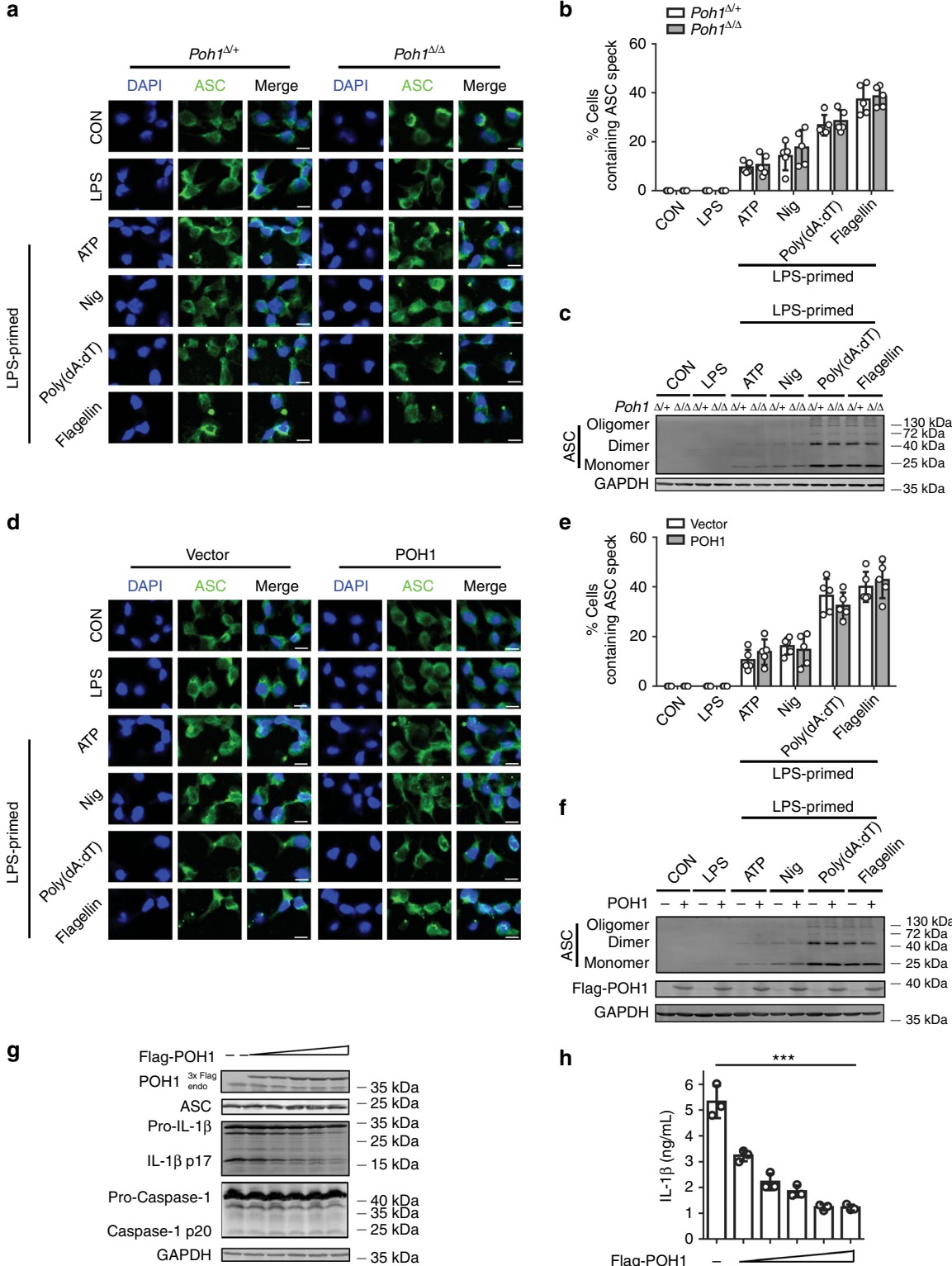

the levels of pro-IL-1β, IL-6 or TNFα transcripts are comparable, suggesting that the NF-κB signalling, which is crucially responsible for activation of these target genes, properly functions in both types of cells. In line with our observation, POH1-depleted murine embryonic fibroblasts (MEFs) are competent in the transcriptional activation of IL-6 and TNFα upon poly(I:C) stimulation[25]. In addition, NF-κB-Luc activation is effectively achieved by IKKβ overexpression in both control and POH1-

depleted HEK293T cells[25]. These data suggest that the POH1-mediated regulation of inflammasome activity or IL-1 production occurs downstream of macrophage priming.

Ubiquitination is a post-translational modification that plays a critical role in regulating inflammasome activity[14]. POH1 functionally modifies its multiple substrates in different scenarios by K63-specific deubiquitination. The interactions of endogenous POH1 with pro-caspase-1, pro-IL-1β and NLRP3 in

**Fig. 4** Effects of POH1 expression on inflammasome-mediated ASC foci formation. **a–c** $Poh1^{\Delta/+}$ and $Poh1^{\Delta/\Delta}$ BMDMs were primed with LPS for 12–14 h and then stimulated with ATP (0.5 h), nigericin (Nig, 0.5 h), poly(dA:dT) (5 h) or flagellin (5 h) as indicated, **a** cells were then fixed and stained for ASC (green) and DNA (blue) (scale bars, 10 μm); **b** percentages of macrophages containing ASC foci were calculated; or **c** cells harvested after different treatments were analysed for ASC polymerization. **d**, **e** Immunofluorescence microscopy of BMDMs transduced with control (Vector) or POH1, the cells were treated as in **a**, **d** and then stained for ASC (green) and DNA (blue) (scale bars, 10 μm); **e** percentages of macrophages containing ASC foci. **f** Immunoblot analysis of ASC polymerization in BMDMs treated as in **d**. **g**, **h** HEK293T cells were transfected with 200 ng of plasmids encoding Flag-pro-caspase-1, Flag-ASC and His-pro-IL-1β with different amounts (31.25, 62.5, 125, 250, 500 ng) of plasmid encoding Flag-POH1 and 500 ng of control plasmid, **g** the cell lysates were immunoblotted with the indicated antibodies; **h** IL-1β levels in supernatants were analysed by ELISA 36 h after transfection. Similar results were obtained from three independent experiments. The results represent the mean ± s.d. of three independent sets of experiments. ***$p < 0.001$ (two-tailed Student's $t$-test)

LPS-stimulated BMDMs suggest that the potential mechanism underlying the POH1 regulation of IL-1β activation may involve deubiquitination of these proteins. A previous study has demonstrated that BRCC3, a K63-specific deubiquitinase, enhances NLRP3 inflammasome activity by deubiquitinating NLRP3[35]. Given that POH1 is a negative regulator of NLRP3 inflammasome activity, POH1 is unlikely to function through regulation of the K63-linked ubiquitination of NLRP3. Indeed, deletion or overexpression of POH1 in BMDMs did not change the efficiency of NLRP3-induced ASC oligomerization. Non-degradative K63-linked polyubiquitination of pro-caspase-1 has been shown to increase the proficiency of pro-caspase-1 cleavage[13]. Our results showed that POH1 expression did not confer appreciable changes in the K63-linked ubiquitin modification and cleavage of pro-caspase-1. Instead, POH1 expression has a significant impact on the K63-linked ubiquitin modification of pro-IL-1β. Interestingly, inflammasome agonists, including ATP, poly (dA:dT) and flagellin, remarkably enhanced the K63-linked ubiquitin modification of pro-IL-1β, indicating that rapid increases in the ubiquitination of pro-IL-1β upon receiving the second signals for inflammasome activation may maximize competence of pro-IL-1β cleavage. Currently, the mechanisms underlying the upregulated ubiquitin modifications of pro-IL-1β is unclear. Potential involvement of E3 ligases in these scenarios may need to be considered and evaluated. Remarkably, TRAF3 has been identified as an E3 ligase for K63-linked ubiquitination of ASC, whereas K63-linked polyubiquitination of pro-caspase-1 is reportedly dependent on the E3 ligase cIAP2[11,13]. It would be interesting to investigate which E3 ligases participate in the regulation of K63-linked polyubiquitinaition on pro-IL-1β. Of note, the interaction between POH1 and pro-IL-1β did not require the presence of the inflammasome agonists. Presumably, the existence of a POH1–pro-IL-1β interaction preceding inflammasome assembly or caspase-1 activation may ensure the time-efficient or prompt regulation of pro-IL-1β deubiquitination. In addition, the processes can be further augmented by increased POH1 expression in macrophages upon certain TLR stimulations.

Emerging evidence suggests that there are different mechanisms underlying ubiquitination-mediated regulation of pro-IL-1β processing[18,19]. Recently, the atypical ubiquitin E2 conjugase UBE2L3 has been demonstrated to promote proteasome-induced degradation of pro-IL-1β, thereby inhibiting mature IL-1β production[18]. We showed that neither K48-linked ubiquitination nor the protein levels of pro-IL-1β were apparently regulated by POH1. Moreover, Duong et al. revealed that K63-linked polyubiquitination at K133 of pro-IL-1β has physiological relevance in regulating the production of mature IL-1β[19]. Our data demonstrated that although K63-linked ubiquitin polyubiquitination at K133 played a critical role in regulating the cleavage of murine pro-IL-1β, K133 mutation did not entirely abolish POH1-mediated inhibition of the cleavage and secretion of IL-1β. These results suggest that modification of K63-linked ubiquitin chains at other conserved lysine residues is likely involved in POH1-mediated regulation of pro-IL-1β processing. Our present study suggests that deubiquitination of

murine pro-IL-1β at K133 and K247 is important for POH1-mediated regulation of IL-1β activation. In addition, the POH1-mediated regulation of IL-1β maturation, unlike the regulatory function of A20[19], is not limited to the context of NLRP3 inflammasome activation but is also efficient at restricting IL-1β secretion in macrophages upon AIM2 or NLRC4 stimulation. The precise mechanism underlying the apparent difference is still elusive. Overall, we have identified the deubiquitinase POH1 as a negative regulator of pro-IL-1β processing that restricts excessive immune responses elicited by inflammasome activation. POH1 upregulation induced by TLR stimulation reflects POH1 being a feedback checkpoint for maintaining immune homeostasis (Fig. 6g). As the regulation by POH1 occurs in the distal end of the signalling of activated inflammasomes, POH1 is a potential therapeutic target for manipulating various inflammasome activation-dependent disorders.

## Methods

**Mice.** *Poh1*-floxed mice were previously described[20]. *LysM-Cre*[+] mice in a C57BL/6J background from Jackson Laboratories were crossed with *Poh1*$^{F/+}$ mice to generate *Poh1*$^{\Delta/\Delta}$ mice. Mice were housed in a specific pathogen-free animal facility, and all mice used in the study were gender- and age matched. All experiments were performed according to National Institutes of Health guidelines and were approved by the Institutional Animal Care and Use Committee of Shanghai Jiao Tong University and Shanghai Cancer Institute. *Poh1*$^{\Delta/+}$ and *Poh1*$^{\Delta/\Delta}$ mice at 6–8 weeks of age were co-allocated in the same cages and used for the studies. No randomization methods were used. For the in vivo LPS challenge, male mice were i.p. injected with 40 mg/kg LPS (*Escherichia coli*, 0111:B4, Sigma-Aldrich) or phosphate-buffered saline (PBS). These mice were monitored for their survival/mortality pattern up to 60 h. Blood was collected at the indicated time by facial vein bleeding, and the serum levels of IL-1β, IL-6 and TNFα were measured by enzyme-linked immunosorbent assay (ELISA). In the alum-induced peritonitis experiments, male mice were i.p. injected with 40 mg/kg of alum crystals (Imject Alum, Thermo Scientific), the peritoneal cavities were washed with 5 mL of PBS and the peritoneal fluids were harvested 12 h post-injection. The protein levels of IL-1β, IL-6 and TNFα were measured by ELISA. The numbers of CD11b[+] Ly6G[+] neutrophils and CD11b[+] Ly6C[+] monocytes in peritoneal fluids were analysed by fluorescence-activated cell sorter (FACS).

**Reagents and primary antibodies.** ATP, LPS (*Escherichia coli*, 0111:B4) and poly (I:C) were from Sigma-Aldrich (St Louis, MO); Imject Alum, DSS and lipofectamine 2000 transfection reagent were from Thermo Scientific; mouse granulocyte/macrophage colony-stimulating factor (GM-CSF) and mouse macrophage colony-stimulating factor (M-CSF) were from PeproTech (Rocky Hill, NJ); LPS (*Escherichia coli*, 0111:B4, LPS-EB Ultrapure), flagellin and poly(dA:dT) were from Invivogen (San Diego, CA). The following antibodies used for immunoblot: POH1 (Cell Signalling Technology, 4791, 1:1000), GAPDH (Santa Cruz, sc-25778, 1:1000), β-actin (Santa Cruz, sc-47778, 1:1000), GSDMD (Santa Cruz, sc-393656, 1:500), α-tubulin (Santa Cruz, sc-69969, 1:1000), mouse caspase-1 p45&p20 (AdipoGen, AG-20B-0042, 1:1000), mouse caspase-1 (Santa Cruz, sc-514, 1:1000), IL-1β (Santa Cruz, sc-12742, 1:50), human caspase-1 p45&p20 (AdipoGen, AG-20B-0048, 1:1000), ASC (AdipoGen, AG-25B-0006, 1:1000), NLRP3 (AdipoGen, AG-20B-0014, 1:1000), HA-tag (Sigma-Aldrich, H9658, 1:5000), Flag-tag (Sigma-Aldrich, F1804, 1:1000), V5-tag (MBL, PM003, 1:1000), His-tag (MBL, PM032, 1:1000), IL-18 (MBL, D047-3, 1:500), Lys48-Specific (Millipore, 05-1307, 1:1000), Lys63-Specific (Millipore, 05-1313, 1:1000) Ubiquitin (epitomics, 6708-1, 1:1000) and IL-1β (R&D, AF-401-NA, 1:2000); The following antibodies used for IP: POH1 (Cell Signalling Technology, 4791, 1:200), mouse caspase-1 (Santa Cruz, sc-514, 1:100), IL-1β (Santa Cruz, sc-12742, 1:100), His-tag (MBL, PM032, 1:200) and V5-tag (MBL, PM003, 1:200). Second antibodies: horseradish peroxidase (HRP)-conjugated goat anti-rabbit (Santa Cruz, sc-2004, 1:10,000), HRP-conjugated goat anti-mouse (Santa Cruz, sc-2005, 1:10,000) and HRP-conjugated mouse anti-goat

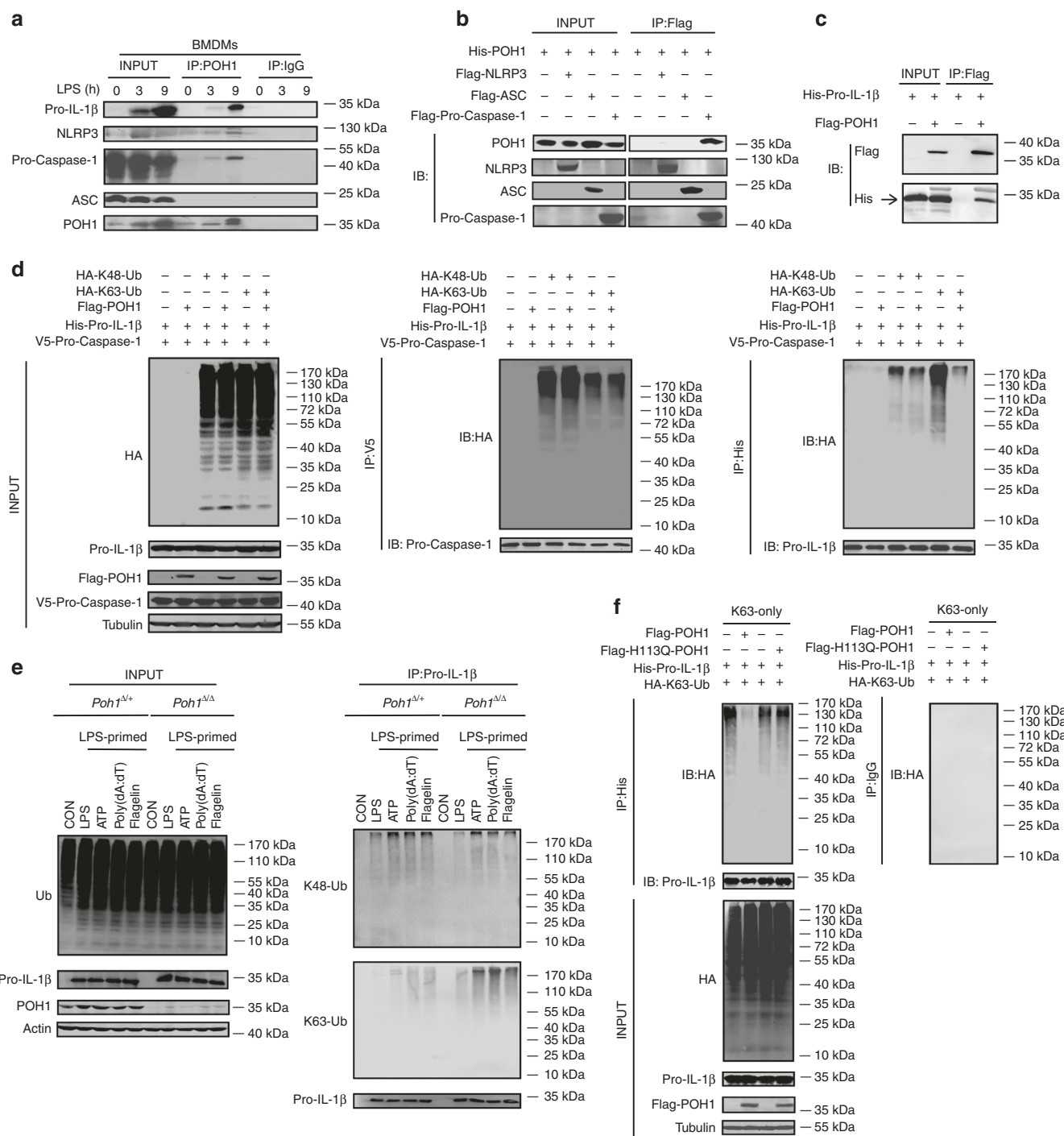

**Fig. 5** POH1 negatively modifies K63-linked polyubiquitination of pro-IL-1β. **a** BMDMs stimulated with LPS for the indicated time periods were immunoprecipitated (IP) with an anti-POH1 or control IgG antibody and immunoblotted (IB) with the indicated antibodies. **b** HEK293T cells transfected with His-POH1 and either Flag-NLRP3, Flag-caspase-1 or Flag-ASC were subjected to IP with an anti-Flag antibody and IB with an anti-His antibody. **c** The lysates of HEK293T cells expressing Flag-POH1 and His-pro-IL-1β were IP with an anti-Flag antibody and IB with an anti-His antibody in cell lysates. **d** HEK293T cells transfected with Vsv-ASC, V5-tagged pro-caspase-1, HA-tagged K48-only or K63-only Ub and His-tagged pro-IL-1β, along with or without Flag-tagged POH1, were subjected to IP with an anti-V5 antibody or anti-His antibody and then IB with the indicated antibodies. **e** Poh1^Δ/+ and Poh1^Δ/Δ BMDMs treated as indicated were subjected to IP with an anti-pro-IL-1β antibody and IB with the indicated antibodies. **f** HEK293T cells were transfected with Flag-caspase-1, Vsv-ASC, His-pro-IL-1β, Flag-POH1 (WT) or Flag-H113Q-POH1, along with HA-tagged K63-only Ub, then cells were subjected to IP with an anti-His or control IgG antibody and IB with the indicated antibodies. Similar results were obtained from three independent experiments

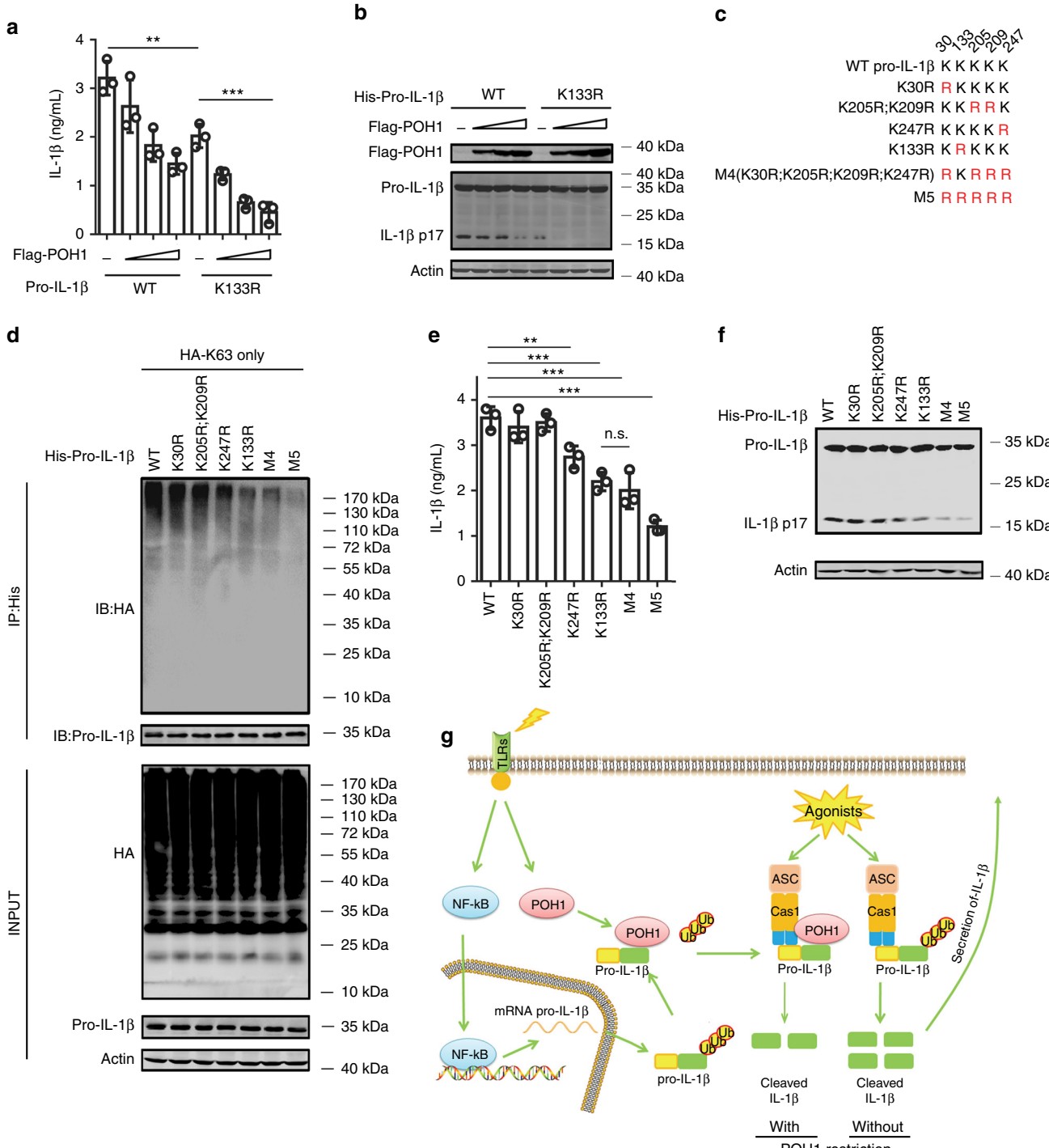

**Fig. 6** Pro-IL-1β maturation is regulated by POH1-mediated deubiquitination. **a**, **b** HEK293T cells were transfected with plasmids encoding Flag-caspase-1, Vsv-ASC, His-pro-IL-1β (WT) or His-pro-IL-1β-K133R with different amounts (125, 250, 500 ng) of plasmid encoding Flag-POH1 and 500 ng of control plasmid, the secretion of **a** IL-1β in the supernatants was quantified by ELISA 36 h after transfection; **b** the cell lysates were IB with the indicated antibodies. **c** Mutants of murine pro-IL-1β with replacement of various lysine residues. **d-f** HEK293T cells were transfected with Flag-POH1, V5-caspase-1, V5-ASC, His-pro-IL-1β (WT) or its mutants, along **d** with or **e-f** without HA-tagged K63-only Ub, then **d** cells were subjected to IP with an anti-His antibody and IB with the indicated antibodies. **e** The IL-1β levels in supernatants were measured by ELISA; or **f** the cell lysates were collected and IB with the indicated antibodies. **g** Proposed function of POH1 as a negative regulator in mediating IL-1β processing. POH1, upregulated by TLR3/4 activation, interacts with and deubiquitinates pro-IL-1β, resulting in restriction of IL-1β cleavage and secretion. Similar results were obtained from three independent experiments. The results represent the mean ± s.d. of three independent sets of experiments. **p < 0.01, ***p < 0.001 (one-way ANOVA with Tukey's post-hoc test)

(Santa Cruz, sc-2354, 1:10,000), goat anti-rabbit IRDye800CW (Li-COR, 926-32211, 1:20,000), goat anti-mouse IRDye800CW (Li-COR, 926-32210, 1:20,000) and donkey anti-goat IRDye800CW (Li-COR, 926-32214, 1:20,000).

**Cell culture**. BMDMs from $Poh1^{\Delta/+}$ and $Poh1^{\Delta/\Delta}$ mice were obtained as described[36]. In brief, bone marrow cells were flushed from the femurs and plated at a density of $4 \times 10^6$/well of a six-well plate in Dulbecco's modified Eagle's medium (DMEM) supplemented with 10% foetal calf serum (FCS), 100 U/mL penicillin, 100 μg/mL streptomycin and 30% L929 culture supernatant or 20 ng/mL GM-CSF (PeproTech). The cells were differentiated for 7 days. HEK293T cells and L929 cells were obtained from American Type Culture Collection (ATCC). Authentication of HEK293T cell line was performed using the GenePrint10 System (Promega Biotech Co.) and via comparisons with the ATCC STR database. L929 cell line, which was not authenticated in our lab, but routinely tested for mycoplasma contamination. Cell lines were cultured at 37 °C in 5% $CO_2$ in DMEM supplemented with 10% FCS, 100 U/mL penicillin and 100 μg/mL streptomycin. BMDMs were primed with 100 ng/mL LPS (*Escherichia coli*, 0111:B4, LPS-EB Ultrapure, Invivogen) for 12–14 h in fresh media. Stimulation with 5 mM ATP (Sigma-Aldrich) or 10 μM nigericin (Invivogen) was performed for 0.5 h after LPS priming. Stimulation with 200 ng/mL poly(dA:dT) or 200 ng/mL flagellin was performed for 5 h by Lipofectamine 2000-mediated transfection after LPS priming. BMDMs were stimulated with 10 ng/mL IL-1β, 10 μg/mL poly(I:C), 100 ng/mL LPS and 5 μM CpG for the indicated times.

**Plasmid transfection and lentiviral infections**. Flag-tagged (3 × Flag) POH1, POH1-H113Q mutant, POH1 with depletion of the JAMM motif (ΔJAMM), carboxyl terminal (ΔN, 181–310 aa) and NH2 terminal (ΔC, 1–180 aa) mutants were cloned into the pLVX (Clontech 632187) vector as described[20]. His-pro-IL-1β, His-pro-IL-1β mutants, V5-ASC, V5-pro-caspase-1 and V5-IL-18 expression plasmids were constructed by PCR-based amplification of complementary DNA from BMDMs and then cloned into the pLVX vector. All constructs were confirmed by DNA sequencing. The Flag-NLRP3, Flag-ASC, Vsv-ASC and Flag-pro-caspase-1 plasmids were provided by Dr. Rongbin Zhou (University of Science and Technology of China). Plasmids expressing HA-tagged K63-ubiquitin (Addgene plasmid 17606) and HA-tagged K48-ubiquitin (Addgene plasmid 17605) were obtained from Addgene (Addgene, Cambridge, MA, USA). Plasmids were transiently transfected into HEK293T cells with Lipofectamine 2000 transfection reagent (Thermo Scientific) according to the manufacturer's instructions. Lentiviral infections were performed using HEK293T cells as producers of viral supernatants. BMDMs were infected with lentivirus on day 3 after thawing in six-well plates. Then, BMDMs were centrifuged at 2500 rpm for 30 min with viral supernatants containing 8 μg/mL polybrene. The viral supernatants were exchanged for fresh media after an overnight incubation. Lentivirus-infected BMDMs were stimulated on day 7 after thawing with the indicated activator.

**Quantitative real-time RT-PCR**. Total RNA of cells or tissues were extracted with the RNAiso Plus kit (Takara Bio Inc.) according to the manufacturer's instructions. Quantitative real-time reverse transcriptase-PCR (RT-PCR) analysis was performed by SYBR Green quantitative PCR kit (Takara Bio Inc.) by using the 7500 Real-Time PCR System (AB Applied Biosystems). The sequences of primers used for RT-PCR are provided in Supplementary Table 1. Data are normalized to GAPDH expression in each sample.

**Immunoblot and immunoprecipitation analysis**. For immunoblot analysis, cells were lysed with RIPA buffer (Thermo Fisher Scientific, 89901) supplemented with a protease inhibitor (Roche Diagnostics, 05892970001) and phosphatase inhibitor cocktail (Roche Diagnostics, 04906845001). The total protein concentrations in the extracts were measured using a BCA protein assay kit (Thermo Fisher Scientific, 23225). Equal amounts of extracts were separated using sodium dodecyl sulfate-polyacrylamide gel electrophoresis (SDS-PAGE) and then were transferred onto NC membranes (Pall Corporation, 66485) for immunoblotting. The blots were incubated with the indicated primary and secondary antibodies and were then visualized by using the SuperSignal West Dura Extended Duration Substrate (Thermo Fisher Scientific, 34076) or an 800 nm laser scanner (Odyssey, Licor, USA). The densitography of the bands were determined by BioRad Image Lab Software or Licor software. For immunoprecipitation analysis, cells were lysed with an immunoprecipitated (IP) lysis buffer (Beyotime Institute of Biotechnology, P0013) supplemented with a protease inhibitor and phosphatase inhibitor cocktail. After centrifugation for 20 min at 12,000 g, supernatants were collected and incubated with 30 μL of Protein G-agarose suspension (Millipore, 16–266) together with specific antibodies (1–2 mg total protein was incubated with 5 μg of indicated antibodies). After 12 h of incubation, beads were washed three times with cold IP buffer. The immunoprecipitates were collected by centrifugation and eluted by boiling with 1% (wt/vol) SDS sample buffer. Uncropped immune blotting results are included in Supplementary Figs. 9-16.

**Immunofluorescence staining and confocal analysis**. For antibody staining, cells were seeded on coverslips and stimulated with the indicated activators. Then, the cells were fixed with methanol and permeabilized with 0.1% Triton X-100 for 10

min at room temperature. After washing three times with PBS buffer, blocking buffer (5% goat serum) was applied for 30 min at room temperature. The cells were incubated with anti-ASC (AdipoGen, AG-25B-0006, 1:200) overnight at 4 °C. After washing three times in PBS, the cells incubated with Alexa Fluor-coupled 488-conjugated or 568-conjugated secondary antibodies (Invitrogen, 1:400) for 1 h. Nuclei were counterstained using DAPI (4′, 6-diamidino-2-phenylindole hydrochloride, Molecular Probes, Invitrogen). The slides were observed under a Leica SP8 confocal microscope (Leica Microsystems, Wetzlar, Germany). For ASC Speck formation, at least 100 cells were counted to determine the percentage of ASC-speck-containing cells.

**ELISA**. The concentrations of mouse IL-1β (88-7013-88, ebioscience), mouse TNFα (88-7324-88, ebioscience), mouse IL-6 (88-7064-88, ebioscience) and mouse IL-18 (7625, MBL) were measured using ELISA kits according to the manufacturer's instructions.

**ASC crosslinking**. BMDMs stimulated with the indicated activator were subjected to crosslinking as described[30]. In brief, the cells were centrifuged and washed three times with PBS buffer. Then, the cells were lysed in 0.5 mL of ice-cold buffer containing 20 mM HEPES-KOH, pH 7.5, 150 mM KCl, 1% Nonidet P-40, protease inhibitor and phosphatase inhibitor cocktail. After shearing 20 times through a 21-gauge needle, the cell lysates were centrifuged at $5000 \times g$ for 10 min at 4 °C and washed three times with PBS buffer. The resultant pellets were resuspended in 1 mL of PBS containing 4 mM DSS for 30 min at room temperature. After centrifugation at $5000 \times g$ for 10 min, the crosslinked pellets were eluted by boiling with 1% (wt/vol) SDS sample buffer and subjected to immunoblot.

**Flow cytometry**. FACS was performed as described[36]. Briefly, cells from the bone marrows and peritoneal cavities were blocked with purified rat anti-mouse CD16/CD32 (553140, BD Biosciences) for 30 min and stained with fluorochrome-conjugated antibodies (CD11b, fluorescein isothiocyanate (FITC)-conjugated, 11-0112-82, eBioscience; Ly6G, PE-conjugated, 551461, BD Biosciences; Ly6C, PerCP-Cy5.5-conjugated, 560525, BD Biosciences). Cells from the spleens were blocked with purified rat anti-mouse CD16/CD32 and stained with fluorochrome-conjugated antibodies (CD11b, FITC-conjugated, eBioscience; Ly6G, PE-conjugated, BD Biosciences; Ly6C, PerCP-Cy5.5-conjugated, BD Biosciences; B220, PE-Cy7-conjugated, 25-0452-82, eBioscience; CD3, PE-conjugated, 553057, BD Biosciences; CD4, PERCP-conjugated, 100432, Biolegend; CD8, PE-conjugated, 12-0081-85, eBioscience; F4/80, PE-conjugated, 12-4801-82, eBioscience; Ly6G, APC-conjugated, 12-5931-82, eBioscience). For PI staining, BMDMs were incubated with PBS containing 0.6 μg/mL PI and then analysed on FACS. Data were acquired by BD Caliber FACS (BD Biosciences). Analysis of the flow cytometry data was performed using Summit 5.2 software. The flow cytometry gating strategies are included in Supplementary Figs. 17.

**Liquid chromatography-electrospray tandem mass spectrometry**. Reversed-phase liquid chromatography (LC)-tandem mass spectrometry (MS/MS) identification of ubiquitinated lysine residues was performed at Applied Protein Technology (PTM Biolabs Inc., Hangzhou, China). In brief, HEK293T cells were transfected with plasmids expressing HA-tagged K63-ubiquitin, Flag-tagged ASC, V5-tagged pro-caspase-1 and His-tagged-pro-IL-1β for 48 h. Cell pellets containing ~$4 \times 10^8$ cells were lysed and isolated pro-IL-1β with an anti-His antibody. The complexes of ubiquitylated pro-IL-1β were separated by 10% SDS-PAGE gel electrophoresis. Coomassie blue-stained bands were excised and destained, followed by reduction and alkylation. After digestion by trypsin for 16 h at 37 °C, samples were desalted with C18 ZipTips and resuspended with 0.1% formic acid (FA) for mass spectrometry analysis.

LC-MS/MS analyses of peptide mixtures were done using the Easy-nLC 1000 Chromatography (Thermo Fisher Scientific) connected to a LTQ Obitrap ETD mass spectrometer (Thermo Fisher Scientific). Samples were separated by nano-flow LC using 150 μm × 100 mm reverse phase 1.8 μm C18 column (Waters) at a flow rate of 300 nL/min. Mobile phase A was 0.1% FA in water, and mobile phase B was 100% acetonitrile containing 0.1% FA. The gradient elution started at 4% of mobile phase B and increased from 4% to 35% during the first 168 min and then increased linearly to 90% of mobile phase B in 4 min.

MS activation type was higher-energy collisional dissociation (HCD) model. MS data were acquired in the m/z range between 350 and 1500 (resolution 30,000). HCD activation time was set to 10 ms and normalized collision energy was set to 35 eV. Raw MS files were analysed by software Proteome Discoverer 1.4. The search parameters were set as follows: Static Modification, Carbamidomethyl (C); Dynamic Modification, Deamidated (N,Q), oxidation (M); Max modification/peptide, 2; Precursor Mass Tolerance, 20 ppm; Fragment Mass Tolerance, 0.05D.

**Statistical analysis**. Statistical comparison was analysed by using the standard two-tailed *t*-test (after the equality of variance was judged by the *F*-test) or one-way analysis of variance (ANOVA) test. The mortality of mice was calculated by the asymmetrical log-rank Mantel–Cox survival test. The data points were not excluded. The researchers involved in this study were not blinded during sample collection or data analysis. Sample sizes were selected based on preliminary results to

ensure a power of 80% with 95% confidence between populations. Values of $p < 0.05$ were considered significant.

## Data availability
The data that support this study are available from the authors upon request.

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

## Acknowledgements
This work was supported by the National Natural Science Foundation of China (81572293, 31770976 and 81672359), the State Key Laboratory of Oncogenes and Related Genes (91-1705 and SB18-08), the Shanghai Sailing Program (no. 18YF1421900), the Shanghai Rising-Star Program (no. 17QA1403700), the Shanghai Municipal Commission of Health and Family Planning (no. 2017YQ040).

## Author contributions
L.Z., Y.L. and Y.Z.L. designed the research, analysed the data and wrote the manuscript; L.Z., Y.L., B.W., G.X., Z.Y., M.T., A.M., T.J. and X.X. performed the experiments; X.Z. contributed reagents and provided intellectual inputs.

## Additional information

**Competing interests:** The authors declare no competing interests.

