## [Peer Review File · Nature Communications]

Reviewers' comments:

Reviewer #1 (Remarks to the Author):

This manuscript reports a novel role for POH1 in de-ubiquitylating precursor IL-1 β to prevent it from being activated by caspase-1. The authors demonstrate that in the absence of POH1, enhanced caspase-1 processing of IL-1 increases IL-1 activation and inflammation both in vitro and in vivo. By and large the results agree with the authors conclusions, although additional controls (indicated below) are required to clarify some of the findings and thereby strengthen the findings.

Figure 3g/h. This data is critical for the manuscripts conclusions. As such, please show more of the western blot membrane for each probing to encompass both pro and cleaved IL-1 and caspase-1 forms (in the supernatants and cell lysates). This is important as it may reveal differences in the levels of cleaved IL-1 forms in the cell lysates, not just supernatants.

Figure 3. The data/conclusions here would be greatly strengthened if the authors could provide western blot time courses examining i) pro-IL-1 levels (following LPS stimulation, with appropriate controls, such as procaspase-1 levels, in WT and POH1 KOs) and ii) activation (e.g. nigericin induced caspase-1 and IL-1 cleavage/secretion in WT and POH1 KOs). Blots for other inflammasome components, such as NLRP3 and ASC, would also be worth performing to ensure specificity of POH1 for IL-1.

Figure 5a. To ensure specificity of the observed interactions please provide the reverse IPs (e.g. IP endogenous pro-IL-1 β /caspase-1 and examine interactions with endogenous POH1). A better control to use in this IP would have been POH1 KO cells.

Figure 5 d. Why isn't caspase-1 detected in the two far left lanes? The blot seems to indicate that these samples have not been transfected with caspase-1 and hence the specificity of this experiment is difficult to assess.

Figure 5e and 5f. Please show input anti-HA (5e) and anti-Ubiquitin (5f) blots –to document transfection/expression levels of Ub in each sample – and hence confirm that the authors conclusions are accurate. Can the authors also blot for K48-Ub (5f) to, again, demonstrate specificity for POH1-mediated K63-de-Ub at the endogenous level?

Figure 5. Please show whole gels for the detection of ubiquitinated proteins in the IP samples (down to at least 10kDa) to help readers to assess the specificity of ubiquitin modifications (i.e. anti His/K63-Ub probings of IP samples; figure 5d-5g).

Figure S3. Does the fact that all the cDNAs of POH1 interacted with IL-1 suggest that the interaction might be non-specific? How common is it for this apparent randomness in a protein-protein interaction?

Figure 6d. Please show input anti-HA (Ub) blots to document expression levels.

Reviewer #2 (Remarks to the Author):

The ms by Liu et al identifies the K68 deubiquitinase POH1 as a negative modulator of IL-1 β cleavage. The work is generally well described, and there is no doubt that POH1 deubiquitinates IL-1 β . However,

there are some problems.

The most general problem is that western blots are never quantified and statistically analyzed. Moreover, gels are too small to see clearly (the white and written areas in the figures can be reduced to have more space for the gels themselves)

The foremost inconsistency is that IL-1b appears to be K68 ubiquitinated on many lysines. This is clear from the fact that the K133R mutation does not preclude the inhibition of IL-1b secretion by POH1 (Fig 6a). Actually, Fig 6b shows that the K133R mutant is more sensitive to POH1 expression and no cleaved IL-1b appears even at the lowest transfection of POH1. How can this be?

Similarly, the pattern of slow-moving bands in Fig 6d is the same in all lanes, whereas the different lanes contain mutations in different lysines, and therefore those lysines cannot be ubiquitinated. One would expect that the bands arising from the ubiquitination of those specific lysines would disappear, but this does not happen. How can this be?

As a minimum, the authors should support their claim that pro-IL1b is K68 ubiquitinated on many lysines with mass spec data.

I am also very disturbed by the fact that in suppl Fig 3 the pattern and intensity of nonspecific bands after IP is the same as the pattern in the input. This would mean that the IP did not work well, and a lot of input material was carried over after IP. How careful were the authors in their IPs? The claim that all fragments of POH1 interact with pro-IL-1b is also weak, given these problems.

Minor problems.

Fig 4 only shows negative results, and should be moved to supplementary figures.

Fig 5 is overdocumented. Since the authors already showed that activation of several TLRs induce the expression of POH1, this info appears redundant here.

Reviewer #3 (Remarks to the Author):

This manuscript by Zhang et al. investigates the function of the deubiquitinase POH1 in regulating inflammasome-dependent immune response. The authors found that POH1-deficiency in the myeloid compartment is sufficient to trigger immunopathology and exacerbates inflammation following microbial or particulate challenge. Mechanistically, POH1 suppresses caspase-1-dependent pro-IL-1b cleavage by removing K63-linked polyubiquitin on pro-IL-1b. These results extend our current knowledge of inflammasome regulation, and should be of the broad interest of the readers of Nature Communications. Majority of the experiments were logical and well performed, however, there are several major questions that need to be clarified before publication.

Major points

1. Mice with myeloid-specific POH1 deficiency developed splenomegaly accompanied with leukophilia. Are serum IL-1b levels already elevated in naïve *Poh1Δ/Δ* animals compared to WT?
2. The authors showed that POH1 does not impact inflammasome assembly but specifically regulates pro-IL-1b cleavage. The authors should also demonstrate POH1-deficiency does not impact the cleavage of other caspase-1 substrate such as GSDMD and IL-18.
3. The authors should also investigate the role of POH1 in driving pyroptosis, so as to strengthen their argument that POH1-deficiency specifically reduces pro-IL-1b cleavage but not cytokine secretion.
4. It is interesting that POH1 interacted with endogenous NLRP3 in BMDM but not HEK-expressed POH1 and NLRP3 did not interact. Could LPS priming promote NLRP3-POH1 interaction?
5. Statistics: It is not clear from the figure legends how many times the animal experiments were repeated and if the data are from one or several repeats.

Minor points

1. Why are LPS and polyIC but not CpG DNA able to trigger POH1 expression? Is POH1 expression TRIF dependent? Since CpG DNA does not induce POH1 expression, it will be of interest to compare IL-1b maturation in CpG DNA-primed WT and Poh1 Δ/Δ cells.
2. How does inflammasome agonist promote pro-IL-1b K63-linked polyubiquitination? This should be discussed.
3. Figure 5d/e: Could the authors provide quantitative data for the WB analysis of K48 and K63 ubiquitination. Also, in Fig 5e, please show that Flag-pro-Casp-1 Ubiquitination does not change in the same assay (as deduced from the figure legends, Flag-pro-Casp-1 and His-pro-II-1b were co-expressed in this experiment, providing the means for a nice internal control).

Reviewer #1 (Remarks to the Author):

This manuscript reports a novel role for POH1 in de-ubiquitylating precursor IL-1 β to prevent it from being activated by caspase-1. The authors demonstrate that in the absence of POH1, enhanced caspase-1 processing of IL-1 increases IL-1 activation and inflammation both in vitro and in vivo. By and large the results agree with the authors conclusions, although additional controls (indicated below) are required to clarify some of the findings and thereby strengthen the findings.

We thank the reviewer for the positive comments and great efforts to improve the quality of our paper. We have addressed each of the specific questions and concerns raised by the reviewer.

1. Figure 3g/h. This data is critical for the manuscripts conclusions. As such, please show more of the western blot membrane for each probing to encompass both pro and cleaved IL-1 and caspase-1 forms (in the supernatants and cell lysates). This is important as it may reveal differences in the levels of cleaved IL-1 forms in the cell lysates, not just supernatants.

Following the suggestion of the reviewer, we have examined the levels of cleaved IL-1 β and caspase-1 in cell lysates using cells stimulated with or without the inflammasome agonists. We found that POH1 deficiency greatly enhanced cleavage of pro-IL-1 β but did not improve cleavage of pro-caspase-1 in cell lysates (labelled as CL). The data have been integrated into **Fig. 3g**. Analyses of cell lysates from cells with or without POH1 overexpression also showed that POH1 could remarkably suppress cleavage of pro-IL-1 β but not of pro-caspase1 in cell lysates (**Fig. 3h**). Thus, the regulation of cleavage of pro-IL-1 β by POH1 can be observed in cell lysates.

2. Figure 3. The data/conclusions here would be greatly strengthened if the authors could provide western blot time courses examining i) pro-IL-1

levels (following LPS stimulation, with appropriate controls, such as procaspase-1 levels, in WT and POH1 KO) and ii) activation (e.g. nigericin induced caspase-1 and IL-1 cleavage/secretion in WT and POH1 KO). Blots for other inflammasome components, such as NLRP3 and ASC, would also be worth performing to ensure specificity of POH1 for IL-1.

We appreciate the reviewer's thoughtful suggestions. We addressed the questions with more detailed experiments. First, we measured protein levels of pro-IL-1 β , NLRP3 and ASC in BMDMs at different time points after LPS stimulation, and the results showed that the expression of these proteins were not significantly affected by POH1 deficiency (**Supplementary Fig. 2g**). Next, we examined possible changes in the activation of NLRP3 inflammasome and pro-IL-1 β cleavage in cells stimulated with ATP or nigericin for different time periods. Our results demonstrate that POH1 deficiency enhances cleavage of pro-IL-1 β without noticeable impact on caspase-1 activation and the expression of NLRP3 and ASC (**Supplementary Fig. 3a, b**).

3. Figure 5a. To ensure specificity of the observed interactions please provide the reverse IPs (e.g. IP endogenous pro-IL-1 β /caspase-1 and examine interactions with endogenous POH1). A better control to use in this IP would have been POH1 KO cells.

Follow the suggestions of the reviewer, we provide new data for supporting the results of **Fig. 5a**. We IP'd endogenous pro-IL-1 β or pro-caspase-1 in BMDMs stimulated with LPS and immunoblotted for POH1, respectively (**Supplementary Fig. 5a**). These results clearly demonstrate that endogenous pro-IL-1 β and pro-caspase-1 interact with POH1.

4. Figure 5 d. Why isn't caspase-1 detected in the two far left lanes? The blot seems to indicate that these samples have not been transfected with caspase-1 and hence the specificity of this experiment is difficult to assess.

We thank the reviewer very much for the correction, and we sincerely apologize for the

mistake we made. In the revised version of our manuscript, we have provided new data regarding to the effects of POH1 on K48- or K63-linked polyubiquitination of pro-caspase-1 and pro-IL-1 β . The impacts of POH1 on ubiquitination of pro-caspase-1 and of pro-IL-1 β are shown in **Fig. 5d middle and right**, respectively, revealing that POH1 can specifically deconjugate K63-linked ubiquitin chains from pro-IL-1 β . The **original Fig. 5d, e** are replaced by **new Fig. 5d** in our revised manuscript.

5. Figure 5e and 5f. Please show input anti-HA (5e) and anti-Ubiquitin (5f) blots –to document transfection/expression levels of Ub in each sample – and hence confirm that the authors conclusions are accurate. Can the authors also blot for K48-Ub (5f) to, again, demonstrate specificity for POH1-mediated K63-de-Ub at the endogenous level?

Following the suggestion of the reviewer, the **original Fig. 5e** showing input of HA-UB expression is replaced with **Fig. 5d (left panel)**. The **original Fig. 5f** is replaced by **Fig. 5e (right panel)**; the input for anti-ubiquitin is shown in **Fig. 5e (left panel)**.

We tested whether K48-linked polyubiquitination of endogenous pro-IL-1 β could be regulated by POH1. In line with previous studies identifying POH1 as a K63-specific deubiquitinase, our results showed that the effect of POH1 expression on K48-linked ubiquitination of pro-IL-1 β is marginal (**Fig. 5e**).

6. Figure 5. Please show whole gels for the detection of ubiquitinated proteins in the IP samples (down to at least 10kDa) to help readers to assess the specificity of ubiquitin modifications (i.e. anti His/K63-Ub probings of IP samples; figure 5d-5g).

Following the suggestions of the reviewer, we have performed new IP experiments by running 10% SDS-PAGE gels to show low-molecular-weight proteins down to 10kDa. The related panels in **Fig. 5** have been updated in new versions.

The **original Fig. 5h** was move to **Supplementary Fig. 6e** due to limited space.

7. Figure S3. Does the fact that all the cDNAs of POH1 interacted with IL-1

suggest that the interaction might be non-specific? How common is it for this apparent randomness in a protein-protein interaction?

The concern of the reviewer is thoughtful. Our co-immunoprecipitation experiments using 293T cell-derived recombinant POH1 showed that wild-type POH1 and POH1 mutants (the Δ JAMM-, Δ C- and Δ N-POH1 mutants) could interact with pro-IL-1 β protein (**original Supplementary Fig 3**). To confirm our previous data, we generated two more truncated POH1 mutants: POH1-M (the mutant form of POH1 with amino acids 141-230) and POH1-C (the mutant form of POH1 with amino acids 231-310). Our IP experiments showed that Δ N-, Δ C-POH1 and POH1-M mutants had the ability to interact with pro-IL-1 β , whereas POH1-C failed to interact with pro-IL-1 β . It seems to us that the region (141-230 AA) of POH1 is critical for the interaction between POH1 and pro-IL-1 β (**Supplementary Fig. 5c, d**).

8. Figure 6d. Please show input anti-HA (Ub) blots to document expression levels.

Since the reviewer 2 recommended us to identify the ubiquitinated lysine sites of pro-IL-1 β with mass spectrometry assays, we re-organized the data of **Fig. 6** in our revised manuscript. The **original Fig. 6** was then replaced with the data using the new constructs of pro-IL-1 β ; some of the results were moved to **Supplementary Fig. 8b**, and the input for HA-Ub expression was included.

Reviewer #2 (Remarks to the Author):

The ms by Liu et al identifies the K68 deubiquitinase POH1 as a negative modulator of IL-1b cleavage.

The work is generally well described, and there is no doubt that POH1 deubiquitinates IL-1b. However, there are some problems.

We thank the reviewer's positive comments on our study, and we very much appreciate the efforts made by the reviewer for improving our paper. We addressed each of the specific questions raised by the reviewer with more data and details.

1. The most general problem is that western blots are never quantified and statistically analyzed. Moreover, gels re too small to see clearly (the white and written areas in the figures can be reduced to have more space for the gels themselves)

Following your kind suggestion, we have provided quantified data of western blots that reach statistical significance in Supplementary Figures for supporting our conclusions, and we have reorganized the panels of the figures to make them more accessible.

2.The foremost inconsistency is that IL-1b appears to be K68 ubiquitinated on many lysines. This is clear from the fact that the K133R mutation does not preclude the inhibition of IL-1b secretion by POH1 (Fig 6a). Actually, Fig 6b shows that the K133R mutant is more sensitive to POH1 expression and no cleaved IL-1b appears even at the lowest transfection of POH1. How can this be?

It is clear that the mutation at K133 site of pro-IL-1 β results in a significant impairment in IL-1 β activation (**Fig. 6a, b**). As the reviewer mentioned, the K133R mutation does not preclude the inhibition of IL-1 β secretion by POH1. The results suggest that other lysine residues on pro-IL-1 β may be also involved in the deubiquitinase activity-dependent effects of POH1 on IL-1 β activation. As shown in **Fig. 6a, b**, the basal levels of released or cleaved K133R IL-1 β are lower than those of WT-IL-1 β . We reason that complete elimination of the ubiquitination of the K133 residue by the mutation might produce additive effects if other lysine residues are subjected to POH-mediated deubiquitination and functionally involved in IL-1 β activation.

To further address the concerns of the reviewer, we did mass spectrometry assays, please find the details below.

3. Similarly, the pattern of slow-moving bands in Fig 6d is the same in all lanes, whereas the different lanes contain mutations in different lysines, and therefore those lysine's cannot be ubiquitinates. One would expect that the bands arising from the ubiquitination of those specific lysines would disappear, but this does not happen. How can this be?

As a minimum, the authors should support their claim that pro-IL1b is K68 ubiquitinated on many lysines with mass spec data.

The suggestion of the reviewer is important. We have performed mass spectrometry analyses to identify the ubiquitinated lysine sites of murine pro-IL-1 β . In brief, we co-expressed His-pro-IL-1 β , Flag-ASC, V5-pro-caspase-1 and HA-K63-only ubiquitin in 293T cells. Then, we isolated pro-IL-1 β with an anti-His antibody and prepared the samples for MS analyses. We found that five lysine residues of murine pro-IL-1 β (K30, K133, K205, K209 and K247) can be ubiquitinated (**Supplementary Fig. 7a-e**). Furthermore, we generated six mutants of pro-IL-1 β to delineate potential contributions of these ubiquitinated lysines on IL-1 β activation. The mutations K133R, K247R or those containing K133R or/and K247R (M4, M5) could remarkably reduce the modification of K63-linked ubiquitination of pro-IL-1 β (**Fig. 6c, d**). Consistently, these mutants (K133R, K247R, M4 and M5) were less efficient in production of released IL-1 β and the cleavage of pro-IL-1 β (**Fig. 6e, f**). In addition, we examined the effects of POH1 expression on the ubiquitination and activation of these pro-IL-1 β mutants (**Supplementary Fig. 8b, c**). Collectively, our results suggest that deubiquitination of murine pro-IL-1 β at K133 and K247 residues is important for POH1-mediated regulation of IL-1 β activation. **Fig 6** in previous version has been updated with new data.

As for the question why the mutations at different lysine residues did not cause apparent changes in the pattern of slow-moving bands or disappear of the bands resulting from ubiquitination, we certainly do not have an exact answer. Using new pro-IL-1 β mutants, we still did not find a remarkable change in the pattern of bands. Interestingly, it was found in recent literatures that mutation of part of ubiquitinated lysine residuals did not cause a significant change in the moving pattern of the proteins examined (for examples, *Nat Immunol.* 2017 Feb;18(2):214-224, Fig6;). We speculate that the moving pattern of ubiquitinated proteins, mostly appearing as a prominent smear, may be not only determined by the number of ubiquitinated lysines but also by the length of conjugated polyubiquitin chain. We think that if multi-lysine residues are ubiquitinated and not completely mutated, the moving pattern of the mutants may be not remarkably changed,

at least for some proteins, owing to the diversities in the length of conjugated polyubiquitin chain and the limits in the techniques (length of gel, visualization with super-amplified signal). Rather, changes in the intensity of bands or smear may be more apparent.

4. I am also very disturbed by the fact that in suppl Fig 3 the pattern and intensity of nonspecific bands after IP is the same as the pattern in the input. This would mean that the IP did not work well, and a lot of input material was carried over after IP. How careful were the authors in their IPs? The claim that all fragments of POH1 interact with pro-IL-1b is also weak, given these problems.

We apologize for the poor quality of the data. We performed new experiments and the original Supplementary Fig. 3 was replaced. In the experiments, we added two more truncated mutants of POH1, and the results showed that while Δ N-, Δ C-POH1 and POH1- M mutants had the ability to interact with pro-IL-1 β , POH1-C failed to interact with pro-IL-1 β . It seems to us that the region (141-230 AA) of POH1 is critical for the interaction between POH1 and pro-IL-1 β (**Supplementary Fig. 5c, d**).

Minor problems.

Fig 4 only shows negative results, and should be moved to supplementary figures.

We agree with the reviewer that most of the data depicted in Fig 4 are negative results. The data are important to emphasize that POH1-mediated restriction of IL-1 β activation does not take place at the processes of inflammasome assembly and activation but at pro-IL-1 β processing. We hope we can keep the data in the main figure that might be easier to be accessed. However, it will not be a problem for us to move them to supplementary figures if it is not suitable.

Fig 5 is overdocumented. Since the authors already showed that activation of several TLRs induce the expression of POH1, this info appears redundant here.

The results present in **Fig. 5** mainly describe the interaction between POH1 and pro-IL-1 β as well as POH1-mediated regulation of pro-IL-1 β ubiquitination.

Reviewer #3 (Remarks to the Author):

This manuscript by Zhang et al. investigates the function of the deubiquitinase POH1 in regulating inflammasome-dependent immune response. The authors found that POH1-deficiency in the myeloid compartment is sufficient to trigger immunopathology and exacerbates inflammation following microbial or particulate challenge. Mechanistically, POH1 suppresses caspase-1-dependent pro-IL-1b cleavage by removing K63-linked polyubiquitin on pro-IL-1b. These results extend our current knowledge of inflammasome regulation, and should be of the broad interest of the readers of Nature Communications. Majority of the experiments were logical and well performed, however, there are several major questions that needs to be clarified before publication.

We very much appreciate the reviewer's positive comments and thoughtful suggestions. We addressed each of the specific questions raised by the reviewer in our revised manuscript with more experiments and details.

Major points

1. Mice with myeloid-specific POH1 deficiency developed splenomegaly accompanied with leukophilia. Are serum IL-1b levels already elevated in naïve *Poh1* Δ/Δ animals compared to WT?

Yes, we have examined serum IL-1 β levels of *Poh1* Δ/Δ mice and their littermates (*Poh1* $\Delta/+$ mice) at 8-12 weeks of age; as shown in **Supplementary Fig. 1d**, the serum levels of IL-1 β were increased in *Poh1* Δ/Δ mice compared to control mice.

2. The authors showed that POH1 does not impact inflammasome assembly but specifically regulates pro-IL-1b cleavage. The authors should also demonstrate POH1-deficiency does not impact the cleavage of other

caspase-1 substrate such as GSDMD and IL-18.

The concerns of the reviewer are thoughtful. To further substantiate our conclusion, we primed BMDMs with or without POH1 deficiency with LPS and subsequently treated them with NLRP3 inflammasome activators ATP or nigericin, AIM2 inflammasome agonist, poly (dA:dT), or NLRC4 inflammasome activator, flagellin, respectively. Compared to their controls, the amounts of IL-18 in supernatants of POH1-deficient BMDMs were not significantly changed (**Supplementary Fig. 4a**). Moreover, POH1 deficiency did not significantly alter the levels of cleaved GSDMD in cell lysates of BMDM stimulated with these agonists (**Supplementary Fig. 4b**). To further confirm our results, we have reconstituted the inflammasome components in HEK293T cells by transient expression of ASC, pro-caspase-1, and their substrates, IL-18 or GSDMD, together with or without POH1. Indeed, POH1-overexpression have little effect on the cleavage of IL-18 or GSDMD (**Supplementary Fig. 4c, d**). Collectively, our results strengthen our conclusion that POH1 regulation of pro-IL-1 β procession is not through modulating caspase-1 activities.

3. The authors should also investigate the role of POH1 in driving pyroptosis, so as to strengthen their argument that POH1-deficiency specifically reduces pro-IL-1b cleavage but not cytokine secretion.

To address this important question, POH1-deficient and control BMDMs stimulated with LPS plus the indicated inflammasome agonists were used for measuring pyroptotic cell death. No significant difference in the inflammasome-dependent pyroptosis was found between POH1-deficient BMDMs and control cells. These results reinforce a specific function of POH1 in regulating pro-IL-1 β cleavage (**Supplementary Fig. 4e**).

4. It is interesting that POH1 interacted with endogenous NLRP3 in BMDM but not HEK-expressed POH1 and NLRP3 did not interact. Could LPS priming promote NLRP3-POH1 interaction?

The question raised by the reviewer is interesting. We agree that LPS priming may possibly promote NLRP3-POH1 interaction. Of note, a previous study reveals that the deubiquitination of K63-linked polyubiquitin modification by the BRCC3 promotes NLRP3-mediated activation of inflammasome and thereby enhances IL-1 β activation

(Mol Cell. 2013 Jan 24;49(2):331-8). In sharp contrast, POH1-mediated deubiquitination restricts pro-IL-1 β cleavage and, more importantly, has no effect on NLRP3-mediated activation of inflammasome. Therefore, we doubt that the interaction between NLRP3 and POH1 is functionally relevant. For these reasons, we did not go further to examine the details of the interaction between POH1 and NLRP3.

5. Statistics: It is not clear from the figure legends how many times the animal experiments were repeated and if the data are from one or several repeats.

We should have described our animal experiments more clearly in the original version of our manuscript. We have updated the legends of **Fig. 1** and **Supplementary Fig. 1** in new version of our manuscript.

Minor points

1. Why are LPS and polyIC but not CpG DNA able to trigger POH1 expression? Is POH1 expression TRIF dependent? Since CpG DNA does not induce POH1 expression, it will be of interest to compare IL-1 β maturation in CpG DNA-primed WT and *Poh1* Δ/Δ cells.

Following the reviewer's comments, we have conducted knockdown experiments. The siRNAs against TRIF and their control were transfected into BMDMs two days before LPS treatment. We found that knockdown of TRIF did not abrogate LPS-induced POH1 expression. The results are shown below.

In addition, we primed BMDMs from *Poh1* Δ/Δ and control mice with CpG for 6 hrs and then treated them with the indicated inflammasome activators. The levels of IL-1 β and its related proteins in the cells lysates or supernatants with different treatments were monitored. The results revealed that the levels of cleaved IL-1 β were significantly increased in POH1-deficient BMDMs, indicating that basal levels of POH1 can regulate pro-IL-1 β cleavage in BMDMs stimulated with CpG and the agonists, though CpG DNA does not induce POH1 expression. The results are shown below.

2. How does inflammasome agonist promote pro-IL-1b K63-linked polyubiquitination? This should be discussed.

The question of the reviewer is thoughtful. As the reviewer mentioned, the inflammasome agonists can promote the K63-linked ubiquitin modification of pro-IL-1 β ; such regulation may maximize competence of pro-IL-1 β cleavage and innate immune responses. Meanwhile, the modification of pro-IL-1 β is tightly controlled by the deubiquitination-related regulatory mechanisms so as to restrain excessive IL-1 β activation.

Interestingly, the stimulation of inflammasome agonists not only upregulates the K63-linked ubiquitin modification of pro-IL-1 β but also of certain inflammasome components. It has been demonstrated that the inflammasome agonists poly (dA:dT) and ATP can rapidly trigger K63-linked ubiquitination of ASC and pro-caspase-1, respectively (Nat Immunol. 2012 Jan 29;13(3):255-63; Immunity. 2011 Dec 23;35(6):897-907). Currently, the mechanisms underlying the upregulated Ub-modifications remain unclear. Potential involvement of E3 ligases in these scenarios may need to be considered and evaluated. Remarkably, TRAF3 has been identified as an E3 ligase for K63-linked ubiquitination of ASC, whereas K63-linked polyubiquitination

of Pro-caspase-1 is reportedly dependent on the E3 ligase cIAP2. (J Immunol. 2015 May 15;194(10):4880-90; Immunity. 2011 Dec 23;35(6):897-907). Therefore, it would be interesting to investigate which E3 ligases participate in the regulation of pro-IL-1 β . Obviously, identification of the E3 ligases that ubiquitinate pro-IL-1 β is critical to fully explore the mechanisms underlying the process of IL-1 β activation. We have integrated these points into the section of “discussion” in the revised manuscript.

3. Figure 5d/e: Could the authors provide quantitative data for the WB analysis of K48 and K63 ubiquitination. Also, in Fig 5e, please show that Flag-pro-Casp-1 Ubiquitination does not change in the same assay (as deduced from the figure legends, Flag-pro-Casp-1 and His-pro-IL-1b were co-expressed in this experiment, providing the means for a nice internal control).

Following the suggestions of the reviewer, we have provided the quantification of K48 and K63 ubiquitination in **Supplementary Fig. 5b, Supplementary Fig. 6a-d and Supplementary Fig. 8a**. In addition, we did more experiments and provided new data to show the ubiquitination of pro-IL-1 β and pro-caspase-1 in the same assay. In line with our previous results, POH1 expression significantly alleviated K63-linked polyubiquitination of pro-IL-1 β but did not affect either K48- or K63-linked polyubiquitination of pro-caspase-1 (**Fig. 5d**).

REVIEWERS' COMMENTS:

Reviewer #1 (Remarks to the Author):

The authors have addressed all my concerns, and I appreciate the efforts they have made to improve the quality of their data and consequently their conclusions. I think their findings will be of great interest to the inflammasome field and readers of Nature Communications.

Reviewer #2 (Remarks to the Author):

The authors have substantially improved their manuscript. The mass experiments and the identification of important ubiquitinated Ks add a lot to the story

Reviewer #3 (Remarks to the Author):

The authors have addressed all my concerns.

Reviewer #1 (Remarks to the Author):

The authors have addressed all my concerns, and I appreciate the efforts they have made to improve the quality of their data and consequently their conclusions. I think their findings will be of great interest to the inflammasome field and readers of Nature Communications.

We are grateful to the reviewer for the positive comments on our revised paper.

Reviewer #2 (Remarks to the Author):

The authors have substantially improved their manuscript. The mass experiments and the identification of important ubiquitinated Ks add a lot to the story.

We appreciate the invaluable suggestions of the reviewer that truly improve our quality of our work.

Reviewer #3 (Remarks to the Author):

The authors have addressed all my concerns.

We thank the reviewer for the positive comments on our revised paper.